# Optimization Algorithms for UAV-and-MUV Cooperative Data Collection in Wireless Sensor Networks

Yu Lu [1,2], Yi Hong [1,2,*], Chuanwen Luo [1,2], Deying Li [3] and Zhibo Chen [1,2]

1 School of Information Science and Technology, Beijing Forestry University, Beijing 100083, China; luyu2000@bjfu.edu.cn (Y.L.); chuanwenluo@bjfu.edu.cn (C.L.); zhibo@bjfu.edu.cn (Z.C.)
2 Engineering Research Center for Forestry-Oriented Intelligent Information Processing of National Forestry and Grassland Administration, Beijing 100083, China
3 School of Information, Renmin University of China, Beijing 100872, China; deyingli@ruc.edu.cn
* Correspondence: hongyi@bjfu.edu.cn

**Abstract:** The deployment of unmanned aerial vehicles (UAVs) has significantly improved the efficiency of data collection for wireless sensor networks (WSNs). The freshness of collected information from sensors can be measured by the age of information (AoI), which is an important factor to consider in data collection. For data collection during long-term mission, the energy limitation of UAVs may cause mission interruption, which makes supplementation of the UAVs' energy more necessary. To this end, we introduce the mobile unmanned vehicle (MUV) to guarantee the UAVs' energy supplementation. In this paper, we investigate the problem of multi-UAVs and single-MUV cooperative trajectory planning (MUSM-CTP) for data collection in WSNs with consideration for the AoI the collected data and the limited battery capacity of UAVs. The objective of this problem is to find cooperative flight trajectories for multiple UAVs and to determine the MUV's travel plan to replace batteries for the UAVs, such that the average AoI of all collected data is minimized. We prove the NP-hardness of the problem and design the algorithm via three phases to solve this: determining candidate hover points based on the affinity propagation (AP) clustering method, constructing the flight trajectories of multiple UAVs based on the genetic algorithm (GA), and designing a travel plan for the MUV. The simulation results verify the effectiveness of the proposed algorithm in improving the freshness of the information collected from all of the sensors.

**Keywords:** unmanned aerial vehicle; wireless sensor network; age of information; mobile unmanned vehicle; cooperative trajectory planning





## 1. Introduction

Wireless sensor networks (WSNs) have been deployed in various domains of human life to provide a wide range of applications, such as intelligent transportation, environmental monitoring, event detection, and target tracking. Generally, the WSN consists of a large number of sensor nodes (SNs) powered by batteries. In traditional WSNs, the information obtained by the sensors is usually collected and forwarded to the destination by receivers in a multi-hop manner, which may quickly deplete the battery of the receiver, greatly reducing the network connectivity, data transmission performance, and the lifespan of the WSN. Additionally, the location of the base station (BS) is limited by geographic conditions such as terrain and altitude. The geographic isolation between the BS and SNs and the complex environment invisibly increases the difficulty and delay of data collection.

As the pace of design and production accelerates, unmanned aerial vehicles (UAVs) have become significant drivers in various fields. In WSNs, UAV-assisted data collection has emerged as a promising method for data collection due to the exceptional flexibility and maneuverability of the UAVs [1]. Compared to conventional ground-based equipment, UAV-assisted data collection exhibits superior environmental adaptability. UAVs act as airborne base stations or relays [2,3]. They approach numerous small, energy-limited SNs

to improve efficiency and coverage. At the same time, the UAVs establish low-altitude and line-of-sight communication links with ground-based SNs to collect perception information from the environment. Each UAV reduces the transmission energy required for data collection via short-range, reliable communication, thus prolonging the lifespan of the entire WSN.

In the UAV-assisted data collection in WSNs, there are two important influencing factors for the network performance: the energy supply of the UAVs and the freshness of the collected information. Firstly, although the utilization of UAVs for data collection in WSNs offers numerous advantages, the limited energy capacity of UAVs imposes an energy constraint on the conduction of long-term and energy-intensive flight missions. Therefore, the energy issue is a fundamental bottleneck in the implementation of UAV-assisted WSNs. To address this challenge, UAVs can supplement their energy supply by replacing onboard batteries by returning to the BS or fixed charging station, which may increase energy consumption and task completion time. To overcome these problems, we attempt a novel approach to enhance the performance of energy-limited UAVs by introducing a mobile unmanned vehicle (MUV) as a mobile charging station for UAVs. The MUV is equipped with a large number of spare batteries and offers the flexibility to compensate for the battery capacity of UAVs while minimizing the impact of onboard energy restrictions. The mobility and speed advantages of the MUV make it possible to perform these replacements quickly and easily. Furthermore, MUV-based energy supplementation is a cost-effective and scalable solution that can be deployed in a distributed manner for simultaneous charging. Secondly, the freshness of information has become increasingly important for applications such as autonomous driving and forest fire monitoring, which can be measured by a new performance metric called the age of information (AoI) [4]. The AoI describes the time elapsed since the latest received update was generated. A lower AoI means that the collected information is more up-to-date, while a higher AoI may be inconsistent with the current state, potentially resulting in the loss of its meaning. Designs based on AoI can ensure the freshness of real-time status information updating systems, which is very different from traditional network designs based on delay and throughput. Therefore, AoI has attracted more and more attention [5–7].

Based on the energy supply of UAVs and the freshness of collected information, this paper considers a new WSN architecture composed of multiple SNs, multiple UAVs, and one MUV. We study the cooperative trajectory planning problem of UAV-and-MUV-assisted data collection in AoI-based WSNs, whose goal is to jointly optimize the flight trajectories of UAVs and the travel plan of the MUV such that the average AoI of the entire WSN is minimized. The contributions of this paper can be summarized as follows:

- We consider a new WSN architecture comprising multiple SNs, multiple UAVs, and one MUV. In this architecture, multiple UAVs are used as mobile data collectors to collect sensed data from SNs deployed on the ground, and an MUV repowers all the UAVs by replacing batteries. We identify the multi-UAVs and single-MUV cooperative trajectory planning (MUSM-CTP) problem, with the goal of minimizing the average AoI of the entire WSN. Then, we prove that this problem is NP-hard.
- In order to solve the MUSM-CTP problem, we propose a three-phase algorithm, the multi-UAVs and single-MUV cooperative trajectory planning algorithm (MUSM-CTPA), which is composed of clustering based on affinity propagation (AP), the UAVs' flight trajectory planning based on the genetic algorithm (GA), and the MUV's travel plan based on the greedy idea.
- We conducted extensive simulation experiments to demonstrate the effectiveness of the proposed algorithm for solving the MUSM-CTP problem.

The remaining parts of this paper are organized as follows. Section 2 introduces the related literature. In Section 3, we introduce related models and the formal definition of our problem. In Section 4, we propose the MUSM-CTPA algorithm to solve the problem. The simulation results are presented in Section 5. Section 6 summarizes this paper.

## 2. Related Works

UAV-assisted WSNs have attracted a lot of attention in both academia and industry. Most existing works have focused on maximizing the UAV's communication coverage range [8], system throughput [9], and energy efficiency [1], or minimizing the UAV's transmission delay [10] and energy consumption [11]. The related works will be introduced from four aspects: resource allocation, path planning, information freshness, and energy supply in UAV-assisted data collection. The comparison and summary of the contributions from related works are presented in Table 1.

**Table 1.** Literature Comparison.

| Related Works | Contributions | | | |
| --- | --- | --- | --- | --- |
| | **Resource Allocation** | **Path Planning** | **Information Freshness** | **Energy Supply** |
| [1,12–17] | $\checkmark$ | [1,12–14] | × | × |
| [18–22] | × | $\checkmark$ | [19] | [18,22] |
| [5,23–27] | [24,26,27] | [24–27] | $\checkmark$ | × |
| [28–34] | [31,32] | [29–32] | × | $\checkmark$ |
| manuscript | $\checkmark$ | $\checkmark$ | $\checkmark$ | $\checkmark$ |

$\checkmark$ indicates presenting features in all literature. × indicates missing features in all literature.

Researchers have extensively studied the efficient mobility management and resource allocation problems in UAV-assisted data collection. In [1], Y. Zeng et al. studied energy-efficient communication between the UAV and ground SNs by optimizing the trajectory of the UAV. A distributed wireless sensor network data collection method is proposed in [12]. The UAVs are used as data transmission nodes to connect separated distributed regions. A UAV-assisted WSN data collection framework is proposed in [13], where UAVs are used as relays to collect sensed data from SNs. In [14], the task completion time is minimized by optimizing the flight path and wake-up time allocation of the UAV. Gong et al. studied the minimization of UAV flight time for collecting sensor data from the starting point in [15]. An HAP-assisted RSMA-enabled VEC system is proposed in [16], where vehicular edge computing (VEC) offers service by utilizing high-altitude platforms (HAPs) and rate-splitting multiple access (RSMA) techniques. In [17], the authors investigated enhancing the service experience of Internet of Things devices (IoTD) by optimizing the limited computation, communication, and energy resources in both HAPs and UAVs when the terrestrial base station (TBS) is underserved.

Trajectory planning is a very important topic in UAV communication consumption. In [18], the cooperative flight trajectories of multiple UAVs are designed by balancing data collection, charging, and collision avoidance. In [19], multiple UAVs act as mobile relay nodes between sensors and base stations. The flight trajectories of multiple UAVs are planned on the premise of minimizing the received packet AoI and equipment energy consumption. In [20], UAV trajectories and SN wake-up scheduling are jointly optimized to minimize the maximum completion time. In [21], a low-complexity algorithm is designed to maximize the total uplink power by jointly optimizing the UAV's position, height, and antenna beam width. The authors of [22] investigated the problem of online cooperative charging schedules using multiple wireless charging vehicles (WCVS) in a WRSN, where each WCV arranges its charging scheduling path by responding to the interdependence of temporal and spatial correlations of different charging requests. The above works focused on optimizing 2D or 3D UAV trajectories and user schedules based on time or energy allocation to improve performance, such as energy efficiency and completion time; however, they ignore the information freshness.

The freshness of sensed information is crucial in delay-sensitive applications. Freshness as an important metric is called the AoI [23], which is defined as the time elapsed since the latest update [5]. Due to the importance of the AoI and the popularity of UAVs in IoT systems, AoI-aware UAV-assisted WSN design has attracted increasing interest. In [24], a centralized information-sharing mechanism among multiple UAVs is designed to avoid

multiple visits to the same user by the UAVs. The AoI is minimized by joint optimization of task allocation, interaction point selection, and UAV trajectory. In [25], the trajectory and data collection mode of the UAVs are jointly optimized to minimize the average AoI of all sensors, where a UAV is allowed to collect data in hover, fly, or mixed modes. In [26], a trajectory planning strategy for UAVs is proposed to minimize the maximum AoI of the WSN by balancing the upload time of sensors and the flight time of the UAVs. In [27], an end-to-end artificial intelligence-based UAV trajectory planning framework is proposed by balancing accuracy and efficiency, and the flight path with the minimum AoI is obtained.

Due to the limited onboard energy of UAVs, it is necessary to replenish energy during their mission. Many researchers have devoted themselves to methods of obtaining energy when UAVs are used as mobile data collectors or mobile edge computing servers. In [28], Suzuki et al. investigated an automatic battery replacement system for UAVs, which included a ground station with replaceable batteries for supplying UAVs. In [29], Luo et al. investigated a laser-charged UAV-assisted wireless rechargeable sensor network. The laser beam directions are uniformly deployed in the monitoring environment to charge the UAV by firing the laser beam. In [30], Fu et al. proposed a reinforcement learning method to design a path for UAVs to collect sensor data and replenish energy using fixed charging stations. In [31], the problem of UAV-assisted data collection in wireless sensor networks with "mobile charging stations" is studied for the first time from the perspective of collaborative trajectory planning. In [32], Zhu et al. investigated the problem of UAV-assisted data collection in large-scale wireless sensor networks based on a truck carrying spare batteries for the UAV's battery replenishment. In [33], the author introduced denial of charge (DoC) into a WRSN and forms an on-demand charging model. A large-scale WRSN with multiple chargers is proposed in [34]; the authors focused on minimizing the charging delay with directional charging schemes.

Since the AoI characterizes the freshness of sensed data in terms of destination, most research works have been devoted to solving the AoI minimization data collection scheduling problem in WSNs. Energy-constrained UAVs need to replenish their energy during the data collection process to continue working. The scheme of returning to the base station or fixed charging station to replace the battery is not only expensive but also increases the energy consumption and task completion time of the UAVs. This is because UAVs need to travel back and forth to the BS regardless of their location, and using fixed charging stations to replace batteries also increases the working time and reduces the flexibility of UAVs. To overcome these problems, this paper studies the collaborative trajectory planning problem of UAV-and-MUV-assisted data collection based on the AoI in WSNs.

## 3. System Model and Problem Definition

### 3.1. System Model

We consider a set of $n$ wireless sensors $S = \{s_1, s_2, \ldots, s_n\}$ located in the two-dimensional region, a set of $m$ UAVs $F = \{f_1, f_2, \ldots, f_m\}$, a charging MUV, and a BS, as shown in Figure 1. In this paper, we use the three-dimensional Cartesian coordinate system XYZ to mark the position of the sensors and UAVs. The sensors are randomly distributed in a given area to monitor the environment. Each SN $s_i \in BS \cup S$ is located at the position of $s_i = (x_i, y_i, 0)$, and $d_{i,j}$ represents the horizontal distance between $s_i$ and $s_j$. Assume that each sensor $s_i \in S$ generates $V_i$ units of sensing data.

We employ $m$ rotary-wing UAVs $F$ as airborne mobile data collectors to collect data from SNs. Each UAV $f_k \in F$ has the same maximum battery capacity $E_{max}$ and minimum energy threshold $E_0$. All UAVs start their data collection task from BS at the same time and fly at fixed height $h$ and constant speed $v$. Let $T$ represent the flight period of the UAVs. At any time, $t \in T$, $f_k$ is located at $pos_k(t) = (x_k(t), y_k(t), h)$, and its remaining energy is denoted by $E_k(t)$. The coverage radius of UAVs when hovering at an altitude $h$ with a transmission range $R$ is $r = \sqrt{R^2 - h^2}$. In this paper, we adopt a one-to-one transmission mode for data collection, where the UAVs can collect data from SNs within the coverage area with a radius $r$ and deliver the data to the BS after completing the data

collection task. The other constraints, such as UAV acceleration or turning angle, are out of consideration here.

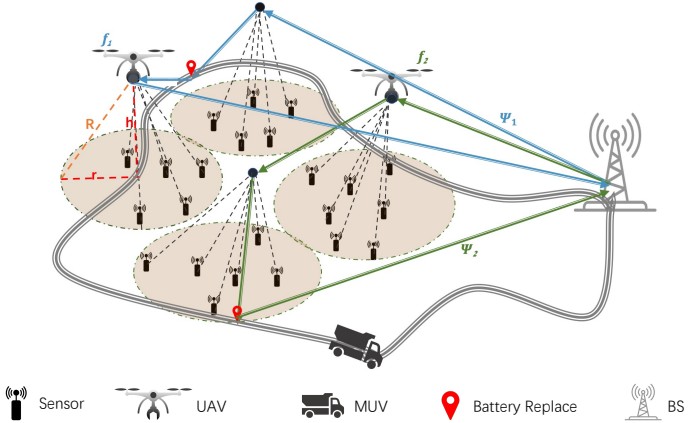

**Figure 1.** System model.

An energy-free MUV is used to replace the batteries of UAVs such that the UAVs have sufficient energy to complete their data collection tasks. The MUV only needs to replace the battery for $f_k$ when the remaining energy $E_k(t)$ is not higher than the energy threshold $E_0$. The MUV starts from BS and travels simultaneously with $F$ along a fixed path $L$. The MUV has a maximum speed $v_{max}$ during the travel process and finally returns to the BS. When a UAV lands on the stationary MUV, the MUV replaces the UAV's battery with a fully charged battery, which takes a fixed amount of time. After the battery replacement, the remaining energy $E_k(t)$ of $f_k$ is set to $E_{max}$. At a particular time, the MUV can replace the battery of only one UAV.

To ensure that a feasible solution exists for the research problem, we assume the following conditions: (1) A UAV with an initial energy level can collect data from at least one area and return to the BS, regardless of where the area is located in the monitored region. (2) The remaining energy of a UAV can support it to fly to the battery replacement point on $L$ and wait for the arrival of the MUV to replace its battery. (3) Any two data collection areas are disjoint from each other. (4) The MUV carries a sufficient number of batteries when it departs from the BS.

*3.2. Data Collection Model*

In order to collect data from the SNs, the UAVs need to hover over the SNs for a period of time. We use $\varpi(s_i)$ to represent a circular data collection area with a radius of up to $r$, whose center is $s_i$. Let $|\varpi(s_i)|$ indicate the number of SNs in area $\varpi(s_i)$. We consider the data transmission between $F$ and $S$ is the line-of-sight communication links (LoS). In one-to-one transmission mode, when $f_k$ hovers directly above $s_i$, $s_i$ is represented as the hover point $c_k^i$ and takes $t_{c,k}^i$ time to collect data from all of the SNs in $\varpi(s_i)$, one by one, in sequence. During this period, the $s_i$ packages its sensing information into a packet with a time stamp $T_i$ and length $L_i$ and then transmits the marked packet to $f_k$. Without losing generality, we assume that F takes off at $T_0 = 0$. The channel power gain of the LoS link from $s_i$ to $f_k$ can be modeled as $g = \beta h^{-2}$, in which $\beta$ represents the channel gain of the reference distance. When $s_i$ uploads data at a constant power $P_i$, its data upload rate on LoS link can be expressed as

$$R_i = B \log_2 \left( 1 + \frac{g \cdot P_i}{\sigma^2} \right) = B \log_2 \left( 1 + \frac{\beta}{h^2 \sigma^2} P_i \right), \tag{1}$$

where $B$ represents the system bandwidth, and $\sigma^2$ represents the noise power of the UAV receiver. Each sensor $s_i$ obtains the sensing data of the $V_i$ unit to be collected. In order

to ensure that $s_i \in \omega(s_i)$ within $t^i_{c,k}$ successfully upload all the data to $f_k$, the following inequality needs to be satisfied.

$$t^i_{c,k} R_i \geqslant \sum_{i=1}^{|\omega(s_i)|} V_i. \tag{2}$$

Compared with the data uploading time of SNs and the flight time of UAVs, it is assumed that the sampling time and communication cost of each sensor can be ignored.

### 3.3. Power Consumption Model

When the UAV performs data collection tasks, both flight and hover operation need to consume energy, and the energy consumption of the UAV mainly consists of these two parts. This paper uses the propulsive power consumption model of the rotary-wing UAV proposed in [10]. Therefore, the power consumption of a UAV flying at a speed of $v$ can be described as

$$P_f = P_0 \left( 1 + \frac{3v^2}{U^2_{tip}} \right) + P_1 \left( \sqrt{1 + \frac{v^4}{4v^4_0}} - \frac{v^2}{2v^2_0} \right)^{\frac{1}{2}} + \frac{d_0 \rho s v^3 A}{2}, \tag{3}$$

where $P_0$ and $P_1$ are two physical constants associated with the UAV and flight environment, $U_{tip}$ represents the tip velocity of the rotor blade, $v_0$ represents the average rotor-induced velocity during hovering, $d_0$ represents the body drag ratio, $s$ represents the rotor stiffness, $\rho$ represents the air density, and $A$ represents the rotor disk area.

Since $v = 0$ when the UAV is hovering, we bring $v = 0$ into Equation (3) and the power consumption of UAV can be expressed as

$$P_h = P_0 + P_1. \tag{4}$$

Let $\tau$ denote the flight period of $f_k$ from the completion of this battery replacement to the next battery replacement, and let $\eta$ denote the set of SNs where $f_k$ collects data in $(0, t)$, where $t \in \tau$. We define the binary variable $X_i(t) \in \{0, 1\}$ for arbitrary $s_i \in S$, $X_i(t) = 1$, which indicates that $s_i$ transmits data to $f_k$ at the time instant $t$, and $X_i(t) = 0$ indicates that $s_i$ does not transmit data to $f_k$ at the time instant $t$. According to the energy change process of $f_k$, the residual energy of $f_k$ at any time $t \in \tau$ can be expressed as

$$E_k(t) = E_{max} - \int_0^t \left( X_i(t) \cdot P_h + P_f \right) dt > 0, \qquad s_i \in \eta. \tag{5}$$

Let $E_{max}$ represent the maximum battery capacity of the UAV. When the current residual energy $E_k(t)$ of $f_k$ is not higher than the energy threshold $E_0$, the MUV shall replace the battery of $f_k$. The UAV battery will be restored to a full charge when it is accessed by the MUV. Therefore, the current residual energy of the $f_k \in F$ must meet $E_k(t) \leqslant E_{max}$.

### 3.4. AoI Model

The freshness of the data collected from the SN is measured by its AoI. Let $\Delta^k_i(t)$ represent the AoI of the data collected by $f_k$ from $s_i$ at a time $t$. According to the definition of AoI, it can be described as

$$\Delta^k_i(t) = \left( t - U^k_i \right)^+, \tag{6}$$

where $1 \leqslant k \leqslant m$, $(a)^+ = max\{0, a\}$. $U^k_i$ is the timestamp when $f_k$ collects data from $s_i$. In fact, when $t < U^k_i$, $s_i$ is not yet served at time $t$, we define $\Delta^k_i(t) = 0$.

Let $U_T^k$ denote the observation time for $f_k$ to transmit the collected data to BS. When $f_k$ returns to BS, the average AoI of all its collected data can be defined as

$$\overline{\Delta_k} = \frac{1}{N(k)} \sum_{i=1}^{N(k)} \Delta_i^k \left( t = U_T^k \right), \tag{7}$$

where $1 \leqslant N(k) \leqslant n$ denotes the number of SN served by $f_k$. The average AoI of the whole network can be defined as

$$\overline{\Delta} = \frac{1}{n} \sum_{k=1}^{m} \sum_{i=1}^{N(k)} \Delta_i^k \left( t = U_T^k \right). \tag{8}$$

### 3.5. Problem Definition

Before giving the formal definition of our problem, we introduce a series of notations: We first use $U_k$ to represent the flight trajectory of $f_k$. During the phase in which $f_k$ collects data from SNs, $C_k$ represents the set of hover points collected by $f_k$ in $U_k$; $T_{C,k}$ represents the set of hover times corresponding to the hover points of $f_k$ in $C_k$. Each hover point $c_k^i \in C_k$ has a corresponding hover time $t_{c,k}^i \in T_{C,k}$. During the phase in which the MUV replaces the battery for $f_k$, $B_k$ represents the set of battery replacement points of $f_k$ in $U_k$; $T_{P,k}$ represents the set of waiting times in $B_k$ for $f_k$ to wait for the MUV to arrive at the corresponding battery replacement point, where $P$ represents waiting for the arrival of the MUV. Each battery replacement point $b_k^l \in B_k$ has a corresponding waiting time $t_{p,k}^l \in T_{P,k}$ and a battery replacement time $t_B$, where $l$ represents the battery replacement point on $L$, and $t_B$ is a fixed value. Then, the flight plan of the UAV $f_k$ is denoted as $\Psi_k(U_k, C_k, T_{C,k}, B_k, T_{P,k})$.

We adopt $\Psi(\Psi_1, \Psi_2, \ldots, \Psi_m)$ to denote the flight plans of $F$, and $\Phi$ to denote the travel plan of the MUV on fixed path $L$. Therefore, $(\Psi, \Phi)$ is a feasible plan for F and the MUV to collect sensed data from all SNs and transmit this to the BS.

In this paper, UAVs are used as mobile data collectors to collect perception data from SNs, while an MUV is used to recharge the UAVs. We try to minimize the impact of onboard energy restrictions on UAV-assisted data collection by giving the charging station mobility to flexibly compensate for the battery capacity of UAVs. Our objective is to find the optimal flight plans for $F$ and the travel plan of the MUV $(\Psi, \Phi)$ cooperative such that the average AoI $\overline{\Delta}$ of all of the collected data is minimized.

We refer to the problem as the multi-UAVs and single-MUV cooperative trajectory planning (MUSM-CTP) problem. Let $r(s_i)$ denote the radius of $\omega(s_i)$. The binary variable $a_{i,j,k}$ is defined as below.

$$a_{i,j,k} = \begin{cases} 1, & \text{if } f_k \text{ visits } \omega(s_j) \text{ after } \omega(s_i), \\ 0, & \text{otherwise.} \end{cases} \tag{9}$$

We can obtain the following mathematical formulation of the MUSM-CTP problem

$$Minimize \quad \overline{\Delta} = \frac{1}{n} \sum_{k=1}^{m} \sum_{i=1}^{N(k)} \Delta_i^k \left( t = U_T^k \right) \tag{10}$$

s.t.

$$\sum_{k=1}^{m} \sum_{j=1}^{n} a_{0,j,k} = m \tag{11}$$

$$\sum_{k=1}^{m} \sum_{i=1}^{n} a_{i,0,k} = m \tag{12}$$

$$1 \leqslant \sum_{k=1}^{m} \sum_{i=1}^{n} a_{i,j,k} \leqslant m \qquad 1 \leqslant j \leqslant n \tag{13}$$

$$0 \leqslant r(s_i) \leqslant r \qquad 1 \leqslant i \leqslant n \tag{14}$$

$$0 < E_k(t) = E_{max} - \int_0^t \left( X_i(t) \cdot P_h + P_f \right) dt \leqslant E_{max} \quad t \in T \tag{15}$$

$$\varpi(s_i) \cap \varpi(s_j) = \varnothing \qquad 1 \leqslant i \leqslant n, \quad 1 \leqslant j \leqslant n \tag{16}$$

$$1 \leqslant N(k) \leqslant n - m + 1 \tag{17}$$

$$X_i(t) \in \{0,1\} \qquad 1 \leqslant i \leqslant n, \quad t \in T \tag{18}$$

Constraints (11) and (12) indicate that each UAV takes off from the BS to reach the data collection area and finally returns to the BS. Constraint (13) indicates that each cluster should be visited by at least one UAV and at most m UAVs. Constraint (14) indicates that the data collection area radius cannot exceed the coverage area radius $r$ of the UAV. Constraint (15) indicates that the energy of all UAVs is always greater than 0 and less than or equal to $E_{max}$. Constraint (16) indicates that any two data collection areas are disjoint from each other. Constraint (17) indicates that each UAV collects data from at least one cluster.

**Theorem 1.** *The MUSM-CTP Problem is NP-hard.*

**Proof.** We consider a special case of the MUSM-CTP problem where we set $V_i = 0$ for each SN $s_i \in S$, $m = 1$, $E_{max} = +\infty$, and $h = 0$. Then, the MUSM-CTP problem can be reduced to the well-known traveling salesman problem (TSP), where the UAVs only need to visit all SNs located in the detection area. Since the TSP problem is proved to be NP-hard, and it is a special case of the MUSM-CTP problem, the MUSM-CTP problem is NP-hard. □

## 4. Proposed Solution

By the definition of MUSM-CTP, we propose an algorithm named the multi-UAVs and single-MUV cooperative trajectory planning algorithm (MUSM-CTPA) to realize cooperative trajectory planning with three phases. In the first phase, we propose a clustering algorithm based on affinity propagation, the minimum coverage algorithm based on affinity propagation (MCA-AP), to divide the SNs into clusters for finding the hovering point set $C$ and the corresponding time set $T_C$ of all UAVs according to the transmission range of the UAV and the location of SNs in the WSN.

Based on the hovering points obtained in the first phase, we will plan the flight trajectories of the UAVs and replace their batteries. In order to efficiently collect data, the separation of UAVs' trajectory planning from UAVs' energy supply is necessary since that UAVs' trajectories are primary for data collection, which is essential for addressing the problem. Furthermore, the MUV travels along the fixed path $L$ and can perform battery replacements for UAVs at any point on path $L$. The candidate battery replacement points are determined based on the energy consumption of UAVs during data collection along the planned trajectories. Then we realize the trajectory planning and energy supply in the following two phases, respectively.

In the second phase, according to the hovering points set $C$ obtained in the first phase, we propose the multi-UAVs trajectory planning algorithm based on the genetic algorithm (MTPA-GA) to design the flight trajectory of multiple UAVs cooperatively traversing all of the data collection areas, which is the positions and order of the hovering points $C_k$ visited

by each UAV $f_k$. Note that each hover point $c_k^i \in C_k$ of $f_k$ is selected from $C$. The algorithm obtains the flight trajectory $U_k$ and the corresponding flight time $T_k$ of each UAV $f_k$.

In the third phase, the MUV will replace batteries at certain locations on the predefined and fixed path $L$ for all UAVs whose energy is not higher than $E_0$ during data collection. According to the flight trajectory $U_k$ of each UAV $f_k$ obtained in the second phase, we calculate the battery replacement point set $B_k$ of each UAV and design the scheduling strategy of all UAVs served by MUV. Then, the corresponding waiting time $t_{p,k}^l \in T_{P,k}$ of each UAV $f_k$ waiting for an MUV at its battery replacement point $b_k^l \in B_k$ is obtained. Finally, the flight trajectories of all UAVs, the travel plan of an MUV, and the time consumption of each UAV to complete the data collection task are obtained, and then the average AoI $\overline{\Delta}$ of all the collected data is obtained. The implementation of the above three phases is detailed in the following three sections.

### 4.1. Algorithm for Minimum Coverage Based on Affinity Propagation

In this subsection, we propose the minimum coverage algorithm based on affinity propagation (MCA-AP) to divide the SNs into clusters to compute the hovering positions and the corresponding hovering time for all UAVs to collect data from all SNs. The SNs randomly deployed in the monitoring area need to be visited by at least one UAV with a limited transmission range for data collection. The MCA-AP aims to divide all SNs in the monitoring area into clusters according to the transmission range of $f_k$, such that the number of divided clusters is as small as possible and the distance between SNs within each cluster is as close as possible. This algorithm identifies a cluster center for the UAV to hover in each cluster, and the UAV hovers above the cluster center to sequentially collect the data of the SNs in this cluster. We apply this algorithm to optimize the number of clusters and obtain the hover point and the corresponding hover time of each cluster.

Before describing the algorithm, we introduce some parameters used in this algorithm. Let $\mathcal{N}$ represent the number of clusters. The binary variable $\rho_{i,j}(i, j = 1, 2, \ldots, n)$ represents whether sensor $s_i$ chooses sensor $s_j$ as its clustering center. When $\rho_{i,j} = 1$, sensor $s_i$ chooses sensor $s_j$ as its clustering center. The variable $r_t(i,j)$ represents the ability of $s_j$ to act as the cluster center of $s_i$ in the $t$-th iteration. The variable $a_t(i,j)$ represents the ability of $s_i$ to select $s_j$ as its cluster center in the $t$-th iteration. The variable $sim(i,j)$ represents the similarity information between $s_i$ and $s_j$.

The MCA-AP consists of the following five steps:

- In the first step, the sample data set $S = \{s_1, s_2, \ldots, s_n\}$ is input. In order to satisfy constraint (14), the coverage radius $r$ of the UAV is input. The counter $num = 0$ is initialized, and the maximum iteration number $N_{max}$ is set to an arbitrarily large positive integer to obtain the best experimental results possible, as shown in Line 1 of Algorithm 1.
- In the second step, based on the sample data set input in the first step, the Euclidean distance $d$ between each sensor is calculated, e.g., $d(i,j)$. Let $-d(i,j)$ initialize the similarity information $sim(i,j)$ between $s_i$ and $s_j$ in the similarity matrix $Sim$. The larger the value of $sim(i,j)$, the closer the distance between $s_i$ and $s_j$, and the stronger the ability of $s_j$ to act as the cluster center of $s_i$, as shown in Line 2–Line 7 of Algorithm 1. Both matrix $R$ and matrix $A$ are initialized as zero matrices isomorphic to matrix $Sim$, where matrix $R$ consists of $r_{t+1}(i,j)$, and matrix $A$ consists of $a_{t+1}(i,j)$, as shown in Line 8 of Algorithm 1.
- In the third step, each iteration updates matrix $R$ and matrix $A$ according to (19) and (21), respectively. In order to avoid oscillations, the attenuation coefficient $\lambda$ is introduced in the update. The updated value of this iteration is set to $\lambda$ times the

updated value of the last iteration, plus $1 - \lambda$ times the updated value of the current iteration, according to (20) and (22), respectively, as shown in Line 11 of Algorithm 1.

$$r_{t+1}(i,j) = \begin{cases} sim(i,j) - \max\limits_{j \neq k}\{a_t(i,k) + r_t(i,k)\}, & i \neq j, \\ sim(i,j) - \max\limits_{j \neq k}\{sim(i,k)\}, & i = j. \end{cases} \quad (19)$$
$$\scriptstyle 1 \leqslant i,j,k \leqslant n$$

$$r_{t+1}(i,j) = \lambda \cdot r_t(i,j) + (1 - \lambda)r_{t+1}(i,j). \quad (20)$$
$$\scriptstyle 1 \leqslant i,j \leqslant n$$

$$a_{t+1}(i,j) = \begin{cases} \min\left\{0, r_{t+1}(j,j) + \sum\limits_{k \neq i,j} \max\{r_{t+1}(k,j),0\}\right\}, & i \neq j, \\ \sum\limits_{j \neq k} \max\{r_{t+1}(k,j),0\}, & i = j. \end{cases} \quad (21)$$
$$\scriptstyle 1 \leqslant i,j,k \leqslant n$$

$$a_{t+1}(i,j) = \lambda \cdot a_t(i,j) + (1 - \lambda)a_{t+1}(i,j). \quad (22)$$
$$\scriptstyle 1 \leqslant i,j \leqslant n$$

- In the fourth step, matrix $R$ and matrix $A$ are used to determine whether there is any change in the decision information of the $t$-th iteration. If there is any change, it will return to the third step to update $r_{t+1}(i,j)$ and $a_{t+1}(i,j)$, as shown in Line 10–Line 16 of Algorithm 1. The algorithm ends if the decision information remains unchanged after several iterations or the algorithm reaches the $N_{max}$. The clustering results and the number of clusters $\mathcal{N}$ are output, and the association relationship $\rho$ between the cluster center and other SNs within the cluster is established in each cluster. The radius $r'$ of each cluster is calculated based on the current clustering results. When $r' > r$, the SNs in the cluster are extracted. Finally, all extracted SNs that do not meet the requirements are re-clustered, which is to return to the second step, as shown in Line 17–Line 23 of Algorithm 1.
- Finally, the cluster center of $\mathcal{N}$ clusters is the data collection hover point set $C$ of UAV $F$. According to the data upload rate $R_i$, the hovering time set $T_C = \{t_c^1, t_c^2, \ldots, t_c^{\mathcal{N}}\}$ corresponding to the hovering point set $C = \{c_1, c_2, \ldots, c_{\mathcal{N}}\}$ is obtained, as shown in Line 24–Line 26 of Algorithm 1. The pseudo-code of the MCA-AP is given in Algorithm 1.

### 4.2. Algorithm for Multi-UAVs Trajectory Planning Based on Genetic Algorithm

In this subsection, we propose the multi-UAVs trajectory planning algorithm based on the genetic algorithm (MTPA-GA) to compute the flight trajectory $U_k$ and the corresponding flight time $T_k$ of each UAV $f_k$. Based on the hover point set $C$ and the corresponding hover time set $T_C$ obtained in the previous section, all the UAVs start from the BS and return to the BS after traversing all hover points. During this process, each hover point can only be visited once. The MTPA-GA is designed to optimize the flight trajectories of multiple UAVs, such that the total mission time $T = \max\{T_k | f_k \in F\}$ of the last UAV to complete the data collection task is as short as possible. The total mission time $T_k$ of $f_k$ is defined as in (23) without considering the battery replacement of $f_k$ and waiting for the MUV, where $L(U_k)$ is the length of the flight trajectory $U_k$.

$$T_k = \frac{L(U_k)}{v} + \sum_{t_{c,k}^i \in T_{C,k}} t_{c,k}^i. \quad (23)$$

---

**Algorithm 1:** Minimum Coverage Algorithm based on Affinity Propagation (MCA-AP)

---

**Input:** the set of $n$ sensors: $S = \{s_1, s_2, \ldots, s_n\}$,
the coverage radius of UAVs: $r$,
the data update rate: $R_i$
**Output:** the hovering point set $C = \{c_1, c_2, \ldots, c_\mathcal{N}\}$,
the hovering time set $T_C = \{t_c^1, t_c^2, \ldots, t_c^\mathcal{N}\}$

1  Set $num=0$ and $N_{max}$ as a sufficiently large integer;
2  **for** *i from 0 to n* **do**
3     **for** *k from 0 to n* **do**
4        $d_{i,j} = \sqrt{(x_i - x_j)^2 - (y_i - y_j)^2}$;
5        $sim(i,j) = -d_{i,j}$;
6     **end for**
7  **end for**
8  Set matrix $R$ and matrix $A$ as zero matrices isomorphic to matrix $Sim$;
9  Calculate $E_{old} = R + A$;
10  **while** *Not convergent and num < $N_{max}$* **do**
11     Calculate messages $r_{t+1}(i,j)$ and $a_{t+1}(i,j)$ for each pair of SNs $s_i$ and $s_j$ according to (19), (20), (21)
      and (22), respectively;
12     $num = num + 1$ and calculate $E_{new} = R + A$;
13     **if** $E_{new} \neq E_{old}$ **then**
14        $E_{old} = E_{new}$ and return to step 10;
15     **end if**
16  **end while**
17  **for** *i from 0 to $\mathcal{N}$* **do**
18     Calculate the radius $r_i$ of each cluster and compare with the given radius $r$;
19     **if** $r_i > r$ **then**
20        Store the $s_i$ in the cluster into the array point[];
21     **end if**
22  **end for**
23  Return to step 2 and update $n$ to $len(point)$;
24  **for** *i from 0 to $\mathcal{N}$* **do**
25     Calculate hover times set $T_C$ corresponding to hover points set $C$ according to $t_c^i = \frac{V_i \cdot |\varpi(i)|}{R_i}$;
26  **end for**

---

We solve the cooperative trajectory planning problem of multiple UAVs based on the genetic algorithm(GA). By coordinating the number of hover points visited by each UAV, the hover points visited by each UAV and their order are obtained. Designing the MTPA-GA requires defining the following elements: initial population, fitness function, selection, crossover and mutation operations, and the termination criterion.

- Initial population: Each cluster obtained in the clustering phase by the MCA-AP is made up of several SNs. It is known that $\mathcal{N}$ is the number of clusters obtained. Each individual $X$ in the initial population is made up of the cluster heads of the same $\mathcal{N}$ clusters but in a different order within each individual $X$. Since multiple UAVs cooperate to perform the data collection task, which is different from the case of a single UAV, each individual needs to maintain two gene segments: the trajectory segment and the break point segment. Therefore, according to the number $m$ of UAVs, we set $m - 1$ breakpoints in each individual to obtain the trajectory of each UAV, as shown in Line 1 of Algorithm 2.

- Fitness function: In particular, a large fitness value implies that an individual has a high probability of being chosen to form part of the next generation. We consider that an individual $X$ following the shorter path has better fitness and that it has a greater chance of being selected in the next generation, as shown in Line 4 of Algorithm 2. The fitness function $F(X)$ is defined as in (24):

$$F(X) = \frac{1}{Z}, \qquad (24)$$

where $X$ represents individual, and $Z$ is the weighted sum of the total flight distance and flight distances balance of all UAVs when individual $X$ is considered. $Z$ is defined as in (25):

$$Z = D_w \cdot \sum_{k=1}^{m} D_k + B_w \cdot B, \tag{25}$$

where $D_w$ and $B_w$ are positive constants. $D_k$ represents the flight distance of UAV $f_k$. $B$ represents the balance of distances for all UAVs. The $D_k$ and $B$ are defined as in (26) and (27), respectively:

$$D_k = \sum_{c_i, c_j \in C} dist(\pi_k(c_i), \pi_k(c_j)), \tag{26}$$

$$B = \frac{D_{max} - D_{min}}{D_{max}}, \tag{27}$$

where $dist(a, b)$ represents the distance between $a$ and $b$. $\pi_k(c_i)$ is a mapping function and means that $c_i$ is the hover point $c_k^j$ visited by $f_k$. $D_{max} = \max\{D_k | f_k \in F\}$ is the longest flight distance of all UAVs in the individual $X$, and $D_{min} = \min\{D_k | f_k \in F\}$ is the shortest flight distance of all UAVs in the individual $X$.

- Selection strategy: We use tournament selection as the selection strategy as it can be implemented efficiently. It also allows for modification of the number of individuals taking part in the selection. We will choose an appropriate size of tournament because, if it is larger, weak individuals will have fewer chances to be selected as they have to compete with stronger candidates, as shown in Line 5 of Algorithm 2.
- Crossover operation: For the crossover operation, we use the order crossover method. It is easy to implement and requires no overhead operations. The individuals winning the tournament selection are used for order crossover. Note that crossover is performed independently in each individual, as shown in Line 6 of Algorithm 2.
- Mutation operation: For the mutation operation, we use the swapping mutation method. Indeed, according to the mutation probability, an individual undergoes a mutation by swapping two randomly chosen SNs, as shown in Line 7 of Algorithm 2.

Finally, when the maximum number of iterations $N_{max}$ is reached, the proposed MTPA-GA terminates, as shown in Line 3–Line 9 of Algorithm 2. We select the individual with the largest fitness function value from the current population, which contains the flight trajectories $U$ of $m$ UAVs that we have obtained. The flight trajectory $U_k$ of each UAV $f_k$ is obtained by sequentially assigning $m$ trajectories to the corresponding UAV according to the $m - 1$ breakpoints in the chosen individual. Then, the flight trajectories $U$ of $m$ UAVs is obtained by the MTPA-GA, and the total mission time $T = \max\{T_k | f_k \in F\}$ of the last UAV to complete the data collection task is obtained, as shown in Line 10–Line 13 of Algorithm 2. The pseudo-code of the MTPA-GA is given in Algorithm 2.

### 4.3. Algorithm for Multi-UAVs and Single-MUV Cooperative Trajectory Planning

In this subsection, we propose the multi-UAVs and single-MUV cooperative trajectory planning algorithm (MUSM-CTPA). Based on the hovering points and the corresponding hovering times obtained in the first phase and the flight trajectory of each UAV obtained in the second phase, the battery replacement point set $B_k$ of each UAV $f_k$ is calculated, and the scheduling strategy for the MUV to replace the battery of all UAVs in the process of all UAVs performing the data collection task is designed. Then, the waiting time set $T_{P,k}$ of each UAV $f_k$ waiting for the MUV is calculated. The MUSM-CTPA aims to optimize the travel plan of the MUV and minimize the $\overline{\Delta}$ of all collected data on the premise that the

total mission time $T_k$ of the last UAV returning to the BS is as short as possible. In this case, the total mission time $T_k$ of the $f_k$ is defined as in (28).

$$T_k = \frac{L(U_k)}{v} + \sum_{t^i_{c,k} \in T_{C,k}} t^i_{c,k} + \sum_{b^i_k \in B_k} t_B + \sum_{t^l_{p,k} \in T_{P,k}} t^l_{p,k}. \tag{28}$$

---

**Algorithm 2:** Multi-UAVs Trajectory Planning Algorithm based on Genetic Algorithm (MTPA-GA)

---

   **Input:** the set of hovering points: $C$,
       the speed of the UAVs: $v$,
       the number of UAVs: $m$
   **Output:** the flight trajectories of $m$ UAVs: $U = \{U_1, U_2, \ldots, U_m\}$,
       the longest mission time: $T$

1 Set counter *num*=0, iteration number $N_{max}$ as a sufficiently large integer and the longest mission time $T = 0$;
2 Create the initial population and set $m - 1$ breakpoints;
3 **while** *num* $< N_{max}$ **do**
4    Evaluate the fitness function using (24);
5    Select parents through the tournament strategy and set the size to an appropriate value;
6    The individuals who win the tournament selection independently perform order crossover;
7    According to mutation probability, an individual mutates by exchanging two randomly selected genes;
8    Generate a new population by replacing old individuals with new ones, and set *num* = *num* + 1;
9 **end while**
10 Obtain the final population and the fitness function value of each individual with $\mathcal{N}$ cluster heads;
11 Choose the individual with the maximum fitness function value as the optimal flight trajectories $U$ of $m$ UAVs;
12 Obtain the flight trajectory $U_k$ consisting of the hovering point $C_k$ of each UAV $f_k$ by sequentially assigning $m$ trajectories to the corresponding UAV according to the $m - 1$ breakpoints in chosen individual;
13 Obtain $T = \max\{T_k | f_k \in F\}$, where $T_k$ is obtained according to (23);
14 Find the flight trajectories $U$ of $F$ and $T$;

---

The MUSM-CTPA consists of the following five steps:

- In the first step, the WSN is divided into $\mathcal{N}$ clusters according to Algorithm 1, and $C = \{c_1, c_2, \ldots, c_{\mathcal{N}}\}$ represents the hover points with the corresponding hover times $T_C = \{t^1_c, t^2_c, \ldots, t^{\mathcal{N}}_c\}$ for the $\mathcal{N}$ cluster, as shown in Line 1 of Algorithm 3.
- In the second step, the flight trajectory $U_k$ of each UAV $f_k$ is calculated according to Algorithm 2 based on the hovering point set $C$ obtained in the first step, as shown in Line 2 of Algorithm 3.
- In the third step, based on the UAVs' flight trajectories obtained in the second step, we firstly calculate the current residual energy $E_k$ of UAV $f_k$ from hovering points $c^i_k$ to $c^j_k$ by (29). If $E_k$ is greater than the energy threshold $E_0$, the $f_k$ has the ability to provide services for the next cluster according to the trajectory $U_k$, as shown in Line 6–Line 8 of Algorithm 3. Otherwise, the UAV arrives at the nearest point $(x_l, y_l)$ on $L$ according to its current position and takes it as the battery replacement point $b^l_k$ to wait for the arrival of the MUV, and its corresponding waiting time is $t^l_{p,k}$, as shown in

Line 9–Line 13 of Algorithm 3. Eventually, the battery replacement point set $B_k$ in the flight trajectory of the UAV $f_k$ is obtained.

$$E_k = E_k - \frac{d_{i,j}}{v} \cdot P_f - t_{c,k}^i \cdot P_h. \tag{29}$$

- In the fourth step, for each battery replacement point $b_k^l \in B_k$, the average AoI $\overline{\Delta_k^l}$ of the data collected by $f_k$ at the current location is calculated, as shown in Line 11 of Algorithm 3. The value of $\overline{\Delta_k^l}$ is an important criterion to decide the MUV service scheduling strategy.
- In the fifth step, we design the MUV service scheduling strategy based on the greedy idea. During the process of the MUV traveling on the fixed path $L$, it may need to replace the batteries of multiple UAVs at a certain time. In this case, the MUV preferentially reaches the battery replacement point with the largest $\overline{\Delta_k^l}$ and replaces the battery for the UAV at that point, as shown in Line 15 of Algorithm 3. This idea enables the UAV with the largest average AoI of the current collected data to remove the waiting time for the MUV to replace the battery as soon as possible, thus minimizing the $\overline{\Delta}$ of the WSN.

Therefore, we can find the flight plans of $F$ and the travel plan of the MUV $(\Psi, \Phi)$. The pseudo-code of the MUSM-CTPA is given in Algorithm 3.

Figure 2 shows the flowchart for the implementation of the MUSM-CTPA algorithm.

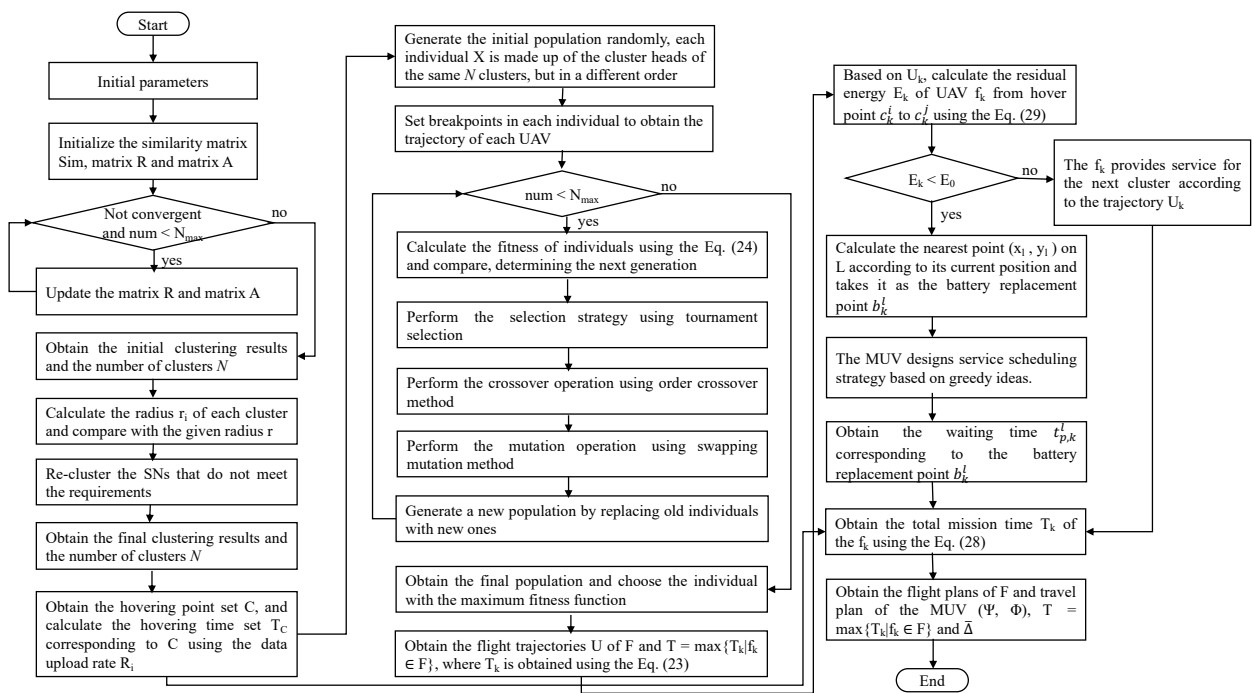

**Figure 2.** The flowchart of the MUSM-CTPA algorithm.

---

**Algorithm 3:** Multi-UAVs and Single-MUV Cooperative Trajectory Planning Algorithm (MUSM-CTPA)

---

**Input:** the set of hovering points: $C$,
the flight trajectories of $m$ UAVs: $U$,
the hovering time set: $T_C$,
the speed of the UAVs: $v$,
the propulsion power of the UAV: $P_f$,
the hovering power of the UAV: $P_h$

**Output:** the flight plans of $F$ and the travel plan of the MUV: $(\Psi, \Phi)$,
the longest mission time: $T$,
the average AoI of the WSN: $\overline{\Delta}$

1   $C = \{c_1, c_2, \ldots, c_\mathcal{N}\}$ and the corresponding time $T_C = \{t_c^1, t_c^2, \ldots, t_c^\mathcal{N}\}$ are obtained by executing Algorithm 1;

2   The flight trajectory $U_k$ of each UAV $f_k$ is obtained by executing Algorithm 2;

3   **for** $k$ *from* 0 *to* $m$ **do**

4     **for** $c_k^i \in C_k$ **do**

5       The residual energy $E_k$ of UAV $f_k$ from hover points $c_k^i$ to $c_k^j$ is calculated by (29);

6       **if** $E_k > E_0$ **then**

7         The UAV $f_k$ serves the next cluster according to the trajectory $U_k$;

8       **end if**

9       **else**

10         The battery replacement point $b_k^l(x_l, y_l)$ on $L$ is calculated;

11         The average AoI $\overline{\Delta_k^l}$ of the collected data flown by the UAV $f_k$ to the current position is calculated;

12         Wait for the arrival of the MUV and set the corresponding waiting time as $t_{p,k}^l$;

13       **end if**

14     **end for**

15     The MUV is based on the greedy idea of preferentially reaching the current battery replacement point with the highest average AoI;

16   **end for**

17   Obtain $T = \max\{T_k | f_k \in F\}$, where $T_k$ is obtained according to (28);

18   Obtain the travel plan $\Phi$ of MUV;

19   Obtain the average AoI $\overline{\Delta}$ of the WSN;

20   Find the flight plans of $F$ and travel plan of the MUV $(\Psi, \Phi)$, $T$ and $\overline{\Delta}$;

---

### 4.4. Computational Complexity

The computational complexity of the proposed MUSM-CTPA algorithm mainly depends on the complexity of Algorithms 1 and 2. Algorithm 1 is based on the AP clustering method to determine the hover points. The computational complexity mainly involves similarity computation and message passing. In the similarity computation stage, the similarity matrix between samples needs to be calculated, which requires a computational cost of $O(N_1^2)$, where $N_1$ is the number of SNs. In the message passing stage, $T_1$ iterations are performed. Therefore, the computational complexity is $O(N_1^2 * T_1)$. Algorithm 2 is based on the GA method to design cooperative flight trajectories for multiple UAVs. The computational complexity depends mainly on the population size, the chromosome length, and the number of iterations. During the initialization stage, generating the initial population typically has a computational complexity of $O(N_2 * L_2)$, where $N_2$ is the population size, and $L_2$ is the chromosome length. Updating the population involves replacing individuals in the old population, with a computational complexity of $O(T_2)$, where $T_2$ is the number of iterations. Therefore, the computational complexity is $O(T_2 * N_2 * L_2)$. In Algorithm 3,

the scheduling strategy for the MUV is designed based on the greedy idea. The MUV needs to select battery replacement points one by one, starting with the one that has the highest average AoI at the current time, until all of the battery replacement points have been served. This process requires $O(N_3^2)$ computational complexity, where $N_3$ is the number of battery replacement points. To this end, the total computational complexity of the proposed MUSM-CTPA algorithm is approximately $O(N_1^2 * T_1 + T_2 * N_2 * L_2 + N_3^2)$.

## 5. Simulation Experiments

### 5.1. Simulation Settings

In this section, we present simulation results to evaluate the performance of the proposed MUSM-CTPA algorithm. The simulation experiments are conducted in Python. In the experiments, we consider the system model shown in Figure 1. We assume that the SNs are deployed in a 1000 m $\times$ 1000 m detection area, the BS is located at the origin $(0,0)$, and UAVs and the MUV depart from BS at time instant $t = 0$. Unless otherwise specified, the major simulation parameters are presented in Table 2.

**Table 2.** Simulation Parameters.

| Notation | Physical Meaning | Values |
|:---:|:---|:---:|
| $h$ | The UAVs' flight height | 50 m |
| $v$ | The UAVs' flight velocity | 20 m/s |
| $v_{max}$ | The MUV's velocity | 10 m/s |
| $r$ | The UAVs' coverage radius | 20 m |
| $B$ | The system bandwidth | 5 MHz |
| $\beta$ | The channel gain at $d_0 = 1$ m | $-60$ dB |
| $P_i$ | The SNs' transmission power | 0.1 watt |
| $\sigma^2$ | The SNs' noise power | $-110$ dBm |
| $E_{max}$ | The UAVs' maximum battery capacity | 10,000 J |
| $E_0$ | The UAVs' minimum energy threshold | 1000 J |
| $V_i$ | The amount of data carried by each sensor $si \in S$ | [100 KB, 200 KB] |
| $L_i$ | The data packet length | 1 Mbits |
| $t_B$ | The fixed battery replacement time | 5 s |
| $P_0$ | Blade power | 14.7517 J |
| $P_1$ | Induced power | 41.5409 J |
| $U_{tip}$ | Tip speed of the rotor blade | 80 m/s |
| $v_0$ | The average rotor-induced velocity | 5.0463 m/s |
| $d_0$ | The fuselage drag ratio | 0.5009 |
| $\rho$ | Air density | 1.225 kg/m$^3$ |
| $s$ | Rotor solidity | 0.1248 |
| $A$ | Rotor disc area | 0.1256 m$^2$ |

We consider the following nine groups of parameter settings, and we create 100 instances to perform the simulation and obtain the average results for each parameter setting: (1) under different algorithms with fixed $n = 300, 600$ and $m = 3$, in terms of the flight trajectories of the UAVs; (2) $n$ varies from 100 to 600 by the step of 100 with fixed $m = 3$ under different algorithms, in terms of the average AoI and the largest mission time; (3) with a fixed $m = 1, 3, 5, 10$ and $n = 300$, in terms of the flight trajectories of the UAVs; (4) $n$ varies from 100 to 600 by a step of 100 with a fixed $m = 1, 3, 5, 10$, respectively, in terms of the average AoI and the largest mission time; (5) $n$ varies from 100 to 600 by a step of 100 with a fixed $m = 1, 3, 5, 10$, respectively, in terms of the travel time of the MUV; (6) $n$ varies from 100 to 600 by a step of 100 with a fixed $m = 3$, with or without an MUV, in terms of the average AoI and the largest mission time; (7) $v$ varies from 10 to 50 by a step of 10 with a fixed $V_{max} = 5, 10, 15$, $m = 3$ and $n = 300$, in terms of the average AoI and the largest mission time; (8) $E_0$ varies from 500 to 2500 by a step of 500 with a fixed $E_{max}$ = 5000, 10,000, 15,000, $m = 3$ and $n = 300$, in terms of the largest mission time; (9) $V_i$ varies from 100 to 500 by a step of 100 with a fixed $m = 1, 3, 5, 10$ and $n = 300$, in terms of the largest mission time.

For evaluation, we use the greedy-based algorithm as a benchmark and compare it with our algorithm in the same system scenario. Both of the algorithms assume that the given location of BS and SNs, the determined hover points at which the UAVs collect data and the replacement points on $L$ at which the MUV performs battery replacement for the underpowered UAVs, can enable the UAVs to complete their missions and return to the BS. The objective of both algorithms is to find the optimal flight plans of $F$ and travel plan of the MUV $(\Psi, \Phi)$ cooperative such that the average AoI $\overline{\Delta}$ of all collected data is minimized.

### 5.2. Simulation Results and Analysis

5.2.1. The Impact of the Parameters of the SNs

Firstly, we investigate the impact of variations in the number of SNs under different algorithms on the flight trajectories of UAVs, we dispatch three UAVs to perform data collection tasks and set the number of SNs to 300 and 600, respectively. Figure 3 shows the flight trajectories of UAVs utilizing two algorithms with varying numbers of SNs when we set $m = 3$. It can be seen that the MUSM-CTPA algorithm results in a shorter total flight distance than the greedy-based algorithm when the number of SNs is 300 and 600, respectively. This is attributed to the GA-based visit sequence optimization algorithm employed by MUSM-CTPA, which enables it to find suboptimal visit sequences capable of escaping local optima and converging to the global optimal solution in a timely manner. In contrast, the greedy-based visit sequence algorithm tends to become trapped in local optima.

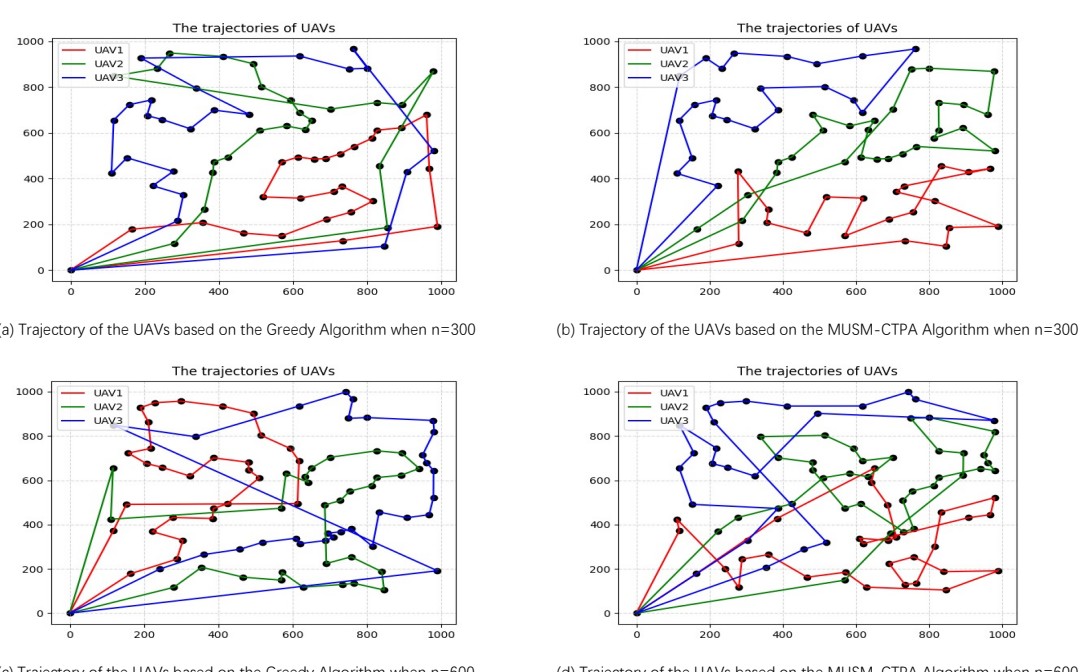

(a) Trajectory of the UAVs based on the Greedy Algorithm when n=300

(b) Trajectory of the UAVs based on the MUSM-CTPA Algorithm when n=300

(c) Trajectory of the UAVs based on the Greedy Algorithm when n=600

(d) Trajectory of the UAVs based on the MUSM-CTPA Algorithm when n=600

**Figure 3.** Illustrate the flight trajectories of three UAVs under different algorithms when $n = 300$ and $n = 600$, respectively.

Figure 4a,b show the impact of varying numbers of SNs on the average AoI and the largest mission time under different algorithms when we set $m = 3$, respectively. It can be seen that both the average AoI and the largest mission time gradually increase in both algorithms as the number of SNs increases. This is because, with an increase in the number of SNs, the number of data collection hover points visited by UAV increases, the number of SNs requiring service at each data collection hover point also increases, and the corresponding hovering time further increases. As a result, the average AoI of all collected data after the UAVs return to the BS increases, while the total mission time of the last UAV returning to the BS becomes longer.

It can also be further seen from Figure 4a,b that regardless of the number of SNs, both the average AoI and the largest mission time based on the MUSM-CTPA algorithm are smaller than those based on the greedy algorithm. This is because the UAV flight trajectory obtained by MUSM-CTPA is more reasonable and closer to the global optimal solution. Therefore, the number of battery replacement times for the UAVs by the MUV will be correspondingly reduced, and the service scheduling of the MUV will be more reasonable, which decreases the average AoI of all the collected data after all UAVs return to the BS and the total mission time after the last UAV returns to the BS.

Figure 5 shows the largest mission time when the amount of data increases from 100 KB to 500 KB by 100 KB under different numbers of UAVs. It can be seen that the largest mission time gradually increases as the amount of data increases under any number of UAVs. However, with any fixed amount of data, the largest mission time gradually decreases with the addition of more UAVs. This is because the corresponding hovering time for each hovering point increases as the amount of data increases, which makes the total mission time of the last UAV returning to the BS become longer. Nevertheless, with the addition of more UAVs, the number of hovering points visited by each UAV decreases, which reduces the total mission time of UAVs.

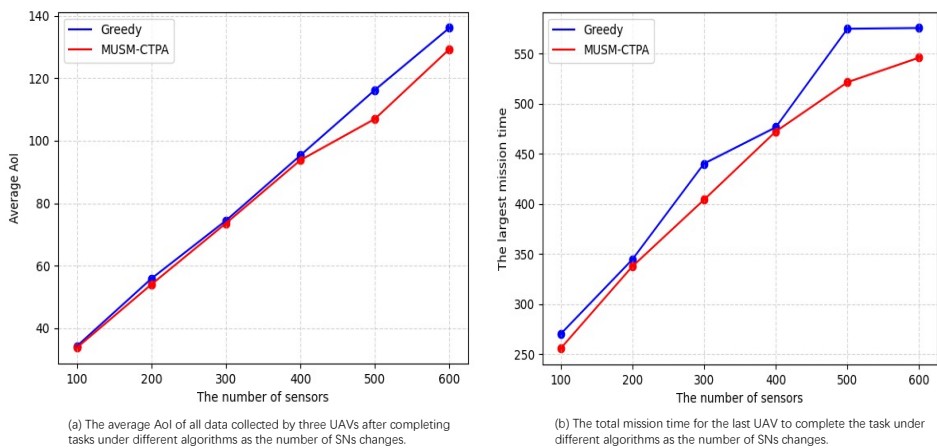

(a) The average AoI of all data collected by three UAVs after completing tasks under different algorithms as the number of SNs changes.

(b) The total mission time for the last UAV to complete the task under different algorithms as the number of SNs changes.

**Figure 4.** The average AoI and the largest mission time under different algorithms versus the number of SNs.

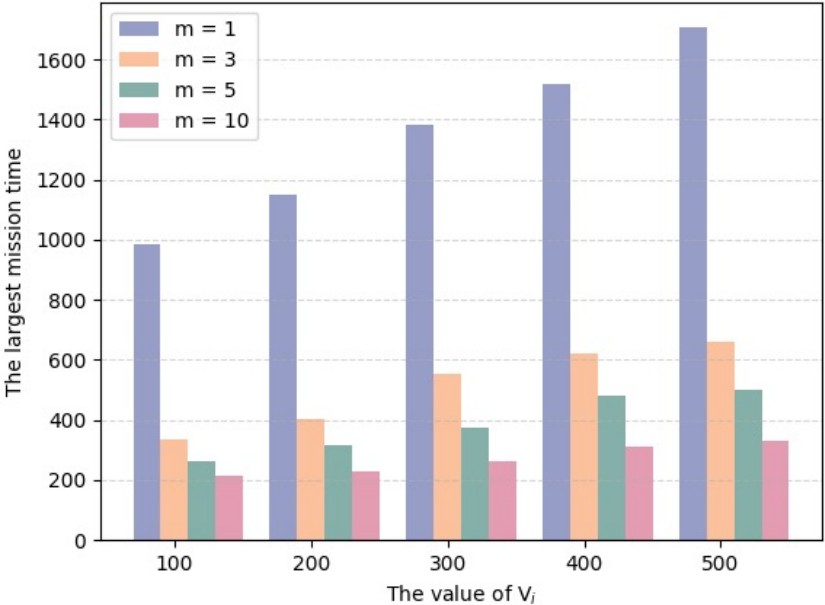

**Figure 5.** The largest mission time under different numbers of UAVs versus the value of $V_i$.

Figure 6 shows the ratio of MUSM-CTPA to the lower bound. It can be seen that the ratio gradually decreases as the number of SNs increases, and the ratio is always greater than or equal to one and less than or equal to two. This is because the MUSM-CTPA algorithm not only plans the flight trajectories of multiple UAVs but also designs a strategy for the MUV to replace the UAVs' batteries to optimize the data collection process. This result validates the effectiveness of MUSM-CTPA in achieving efficient data collection.

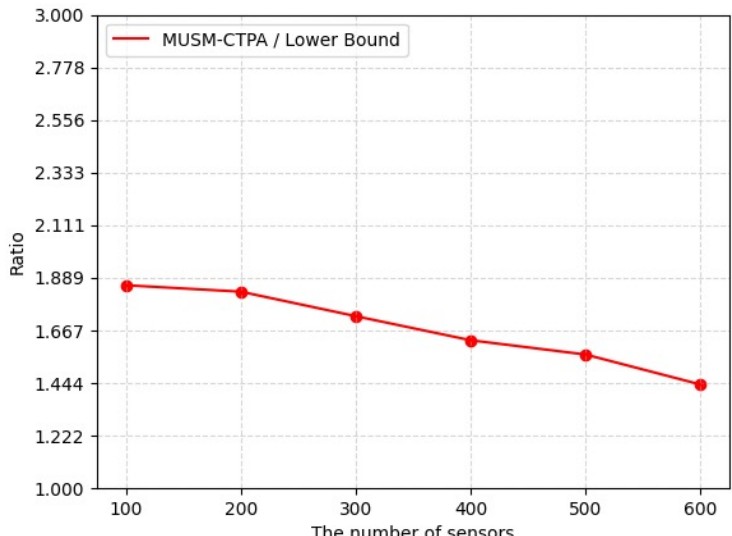

**Figure 6.** The ratio between MUSM-CTPA and the lower bound versus the number of SNs.

### 5.2.2. The Impact of the Parameters of the UAVs

Figure 7 shows the trajectories of the UAVs under the condition of $n = 300$ when we set $m = 1$, $m = 3$, $m = 5$ and $m = 10$, respectively. It can be seen from Figure 7a, when $m = 1$, one single UAV is dispatched to visit all data collection hover points for data collection. It can also be seen from Figure 7b–d that multiple UAVs are dispatched to perform data collection, among which $m = 3$ is set in Figure 7b, $m = 5$ in Figure 7c, and $m = 10$ in Figure 7d. The trajectory of each UAV is marked with a different color. In order to visit all the data collection hover points, the multiple UAVs are supposed to perform data collection collaboratively. Each UAV is responsible for a part of all the data collection hovering points; the balance of the flight distances of all the UAVs should be considered to avoid a much larger gap at the same time. Therefore, the flight distance and mission time of each UAV decrease as the number of UAVs increases.

Figure 8 shows the interaction between the UAVs and one MUV under the condition of $n = 300$ when we set $m = 1$ and $m = 3$, respectively. We use a blue star to indicate the battery replacement point. It can be seen that the MUV needs to replace the battery for the UAV in both Figure 8a,b. In Figure 8a, since there is only one UAV, the MUV only needs to follow the flight of the UAV to provide service in time. In Figure 8b, since there are three UAVs performing tasks at the same time, the MUV needs to adopt a scheduling strategy to decide the service order when replacing the batteries for the UAVs. It can also be seen that there is not much difference in the total number of battery replacements in Figure 8a,b. This is because, compared with one UAV, multiple UAVs performing tasks at the same time may have a longer total flight distance, but the task execution time is much shorter.

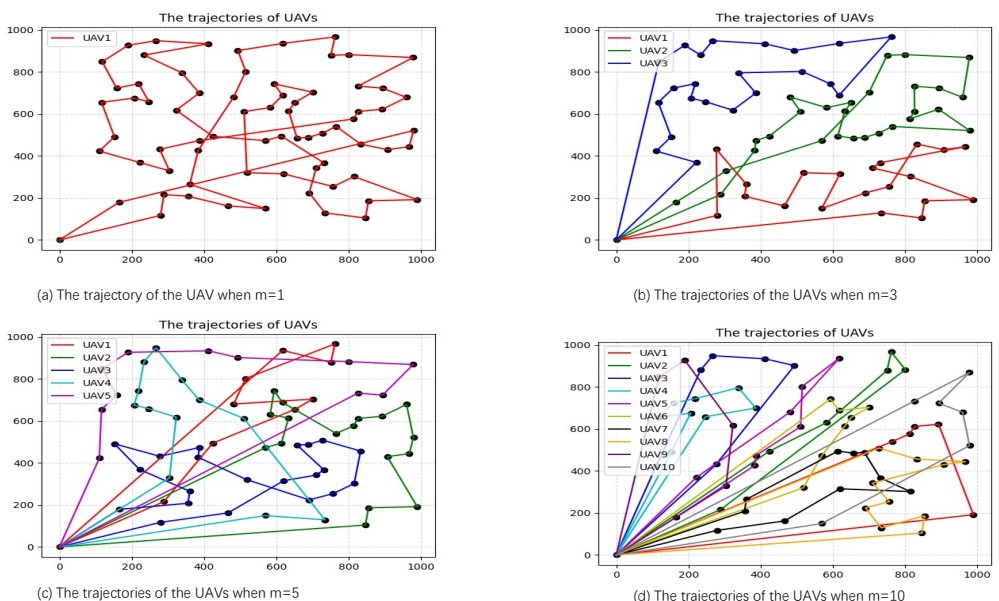

**Figure 7.** Illustration of the trajectories of UAVs when $m = 1$, $m = 3$, $m = 5$, and $m = 10$ under the condition of $n = 300$.

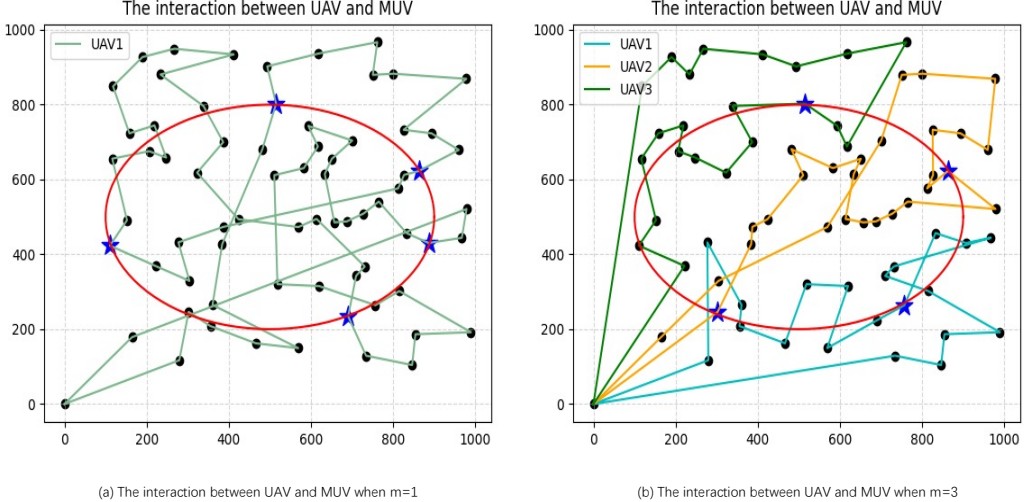

**Figure 8.** Illustration of the interaction between UAV and MUV when $m = 1$ and $m = 3$ under the condition of $n = 300$.

Figure 9a,b show the average AoI and the largest mission time when the number of SNs increases from 100 to 600 by 100 under different numbers of UAVs, respectively. It can be seen that both the average AoI and the largest mission time gradually increase as the number of SNs increases under any number of UAVs. This is because, as the number of SNs increases, the number of data collection hover points visited by UAV increases. The number of SNs required to be served at each data collection hover point also increases, and the corresponding hovering time further increases, which will increase the average AoI of all collected data after the UAVs return to the BS, and the total mission time of the last UAV returning to the BS becomes longer.

It can also be further seen from Figure 9a,b that regardless of the number of SNs, both the average AoI and the largest mission time when $m = 1$ are much higher than the other three multi-UAVs cases. This is because multiple UAVs can perform data collection tasks at the same time, which can complete data collection faster than a single UAV. At the same time, multiple UAVs can work together to assign tasks according to the requirements of data collection tasks, thus covering a wider area and improving the coverage and efficiency

of data collection. Therefore, as the number of UAVs increase, both the average AoI of all the collected data and the total mission time for the last UAV to complete the task decrease correspondingly.

Figure 10 shows the largest mission time under $E_{max}$ and $E_0$ change when we set $n = 300$ and $m = 3$, respectively. It can be seen that the largest mission time gradually increases with the increase in $E_0$ under $E_{max} = 5000$ and $E_{max} = 5000$. However, the largest mission time remains unchanged when $E_{max} = 15,000$. This is because, with the increase in $E_0$, the UAVs need to replace the battery more times, which makes the largest mission time increase. However, once the $E_{max}$ reaches a certain value, the UAVs are fully capable of collecting data without having to replace the battery.

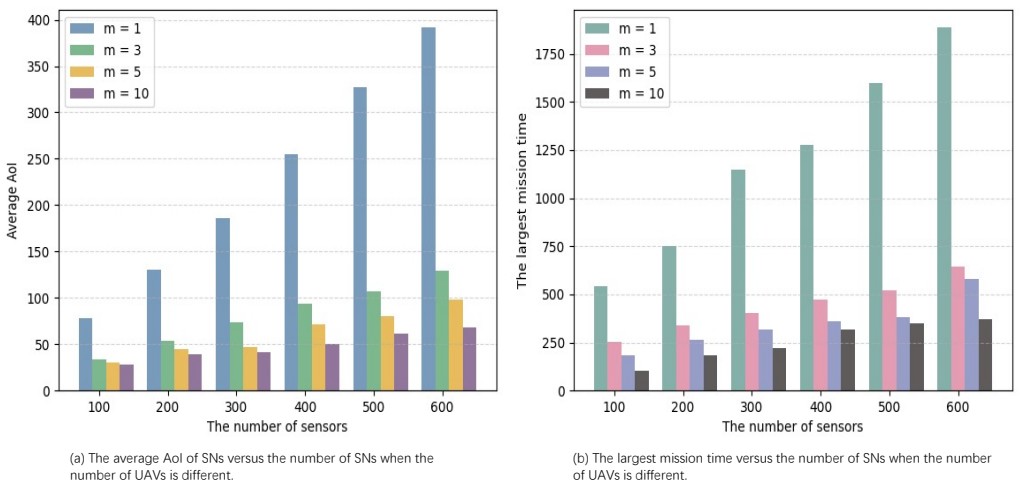

(a) The average AoI of SNs versus the number of SNs when the number of UAVs is different.

(b) The largest mission time versus the number of SNs when the number of UAVs is different.

**Figure 9.** The average AoI and the largest mission time under different numbers of UAVs versus the number of SNs.

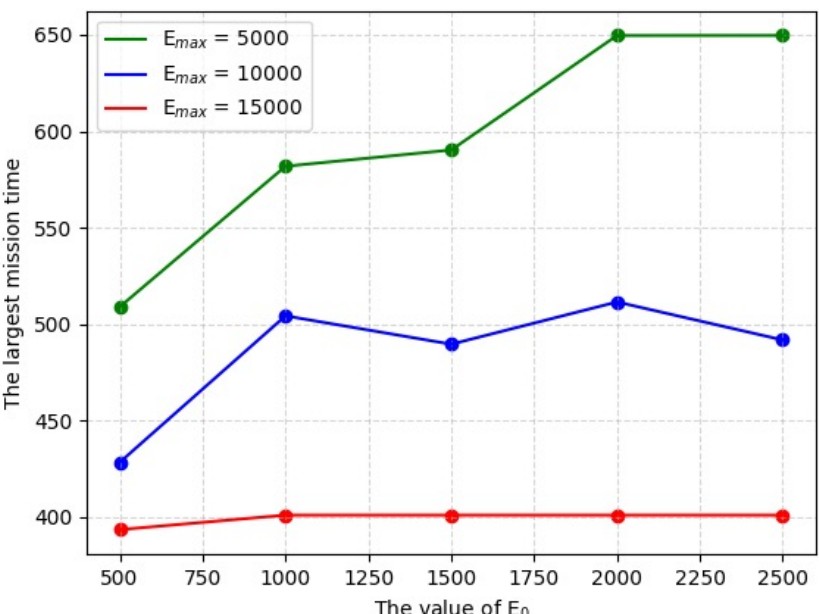

**Figure 10.** The largest mission time versus different $E_{max}$ and $E_0$.

### 5.2.3. The Impact of the Parameters of the MUV

Figure 11 shows the travel time of the MUV returning to the BS after departing from the BS to serve all the UAVs requiring battery replacement when the number of SNs increases from 100 to 600 by 100 under different numbers of UAVs. It can be seen that with any number of UAVs, the travel time of the MUV gradually increases with the increase

in the number of SNs. However, with any number of SNs, the travel time of the MUV gradually decreases with the increase in the number of UAVs. This is because in the case of a given number of UAVs, as the number of SNs increases, the number of data collection hover points visited by UAVs increases, resulting in a greater energy consumption of the UAVs. Consequently, the frequency of battery replacement for the UAVs increases, and the scheduling of the MUV on the fixed path $L$ becomes more frequent. As a result, the travel time of the MUV also increases. However, in the case of a certain number of SNs, multiple UAVs cooperate to visit all data collection hovering points, and the energy consumption of UAVs is reduced. Consequently, the frequency of battery replacement for the UAVs decreases. As a result, the travel time of the MUV also decreases.

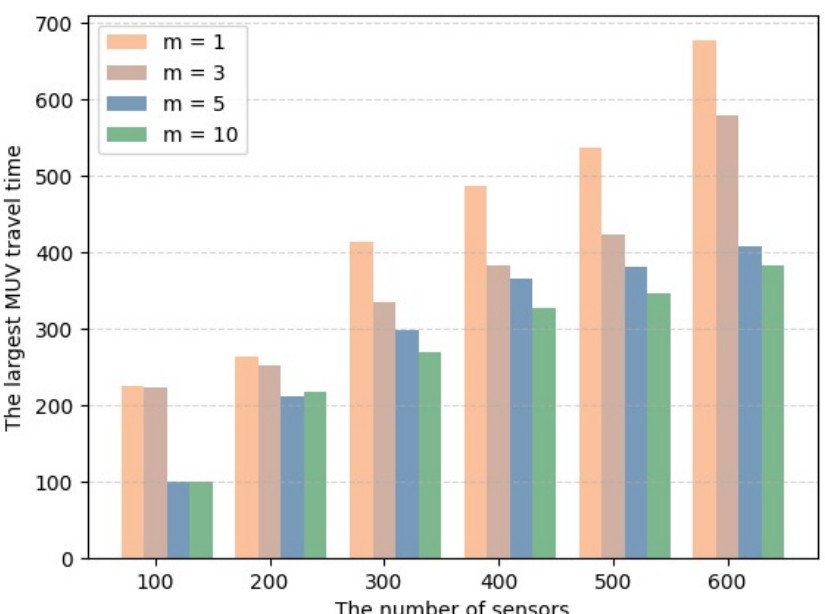

**Figure 11.** The travel time of the MUV under different numbers of UAVs versus the number of SNs.

Figure 12a,b show the average AoI and the largest mission time with or without an MUV to replace the battery for the UAVs when we set $m = 3$, respectively. It can be seen that both the average AoI and the largest mission time with MUV are smaller than those without the MUV. This is because, with the assistance from the MUV, the UAVs can arrive at the nearest point on $L$ to the current position and wait for the MUV to arrive to replace its battery. When there is no MUV, the UAVs need to return to the BS for battery replacement, which greatly increases the distance and flight time of the UAVs.

Figure 13a,b show the average AoI and the largest mission time under $v_{max}$ and $v$ change when we set $n = 300$ and $m = 3$, respectively. It can be seen that both the average AoI and the largest mission time significantly decrease with the increase in $v$ under any $v_{max}$, but eventually flattens out. This is because, on the one hand, with the increase in the UAVs' speed $v$, the flight times of UAVs are shortened, which makes UAVs return to the BS faster. On the other hand, the increase in the MUV's speed $v_{max}$ makes the MUV provide battery replacement service for the UAV faster, which reduces the time for UAVs to wait for the MUV.

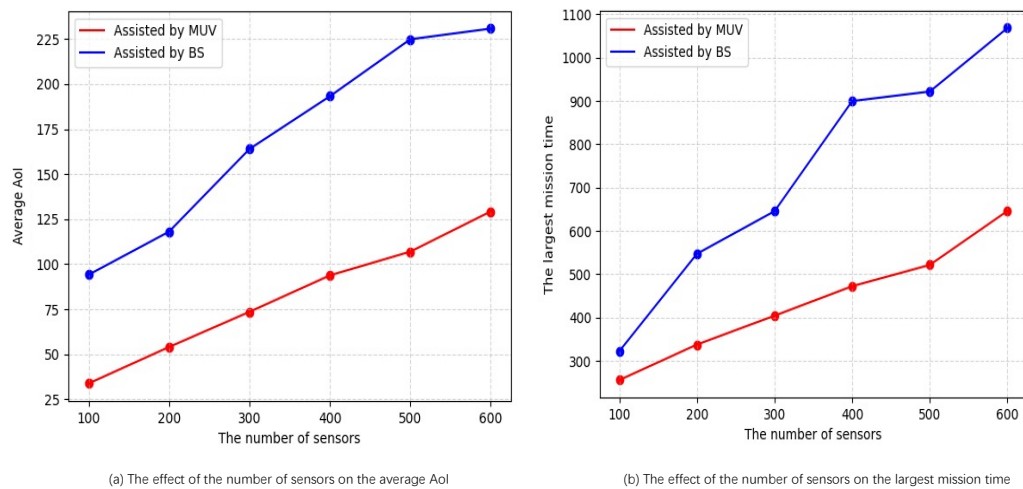

(a) The effect of the number of sensors on the average AoI　　　(b) The effect of the number of sensors on the largest mission time

**Figure 12.** The average AoI and the largest mission time versus with or without a MUV.

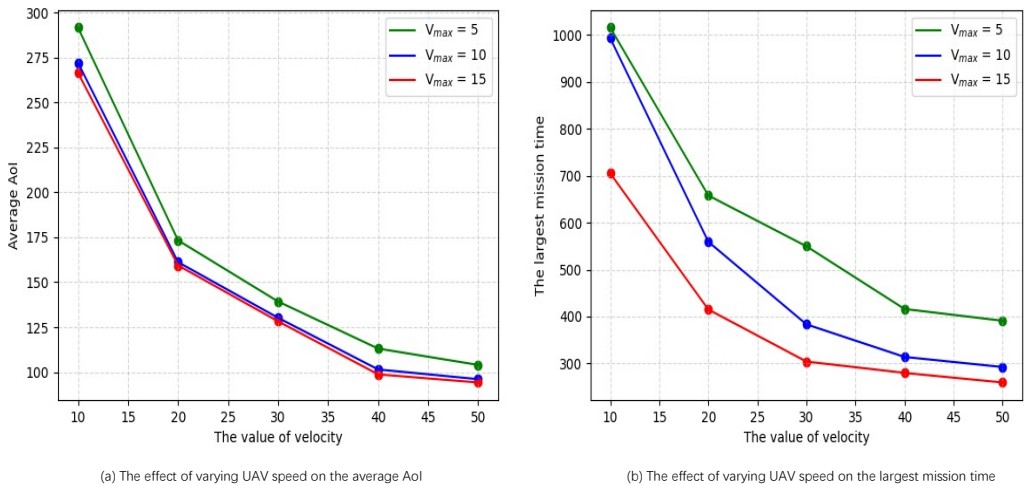

(a) The effect of varying UAV speed on the average AoI　　　(b) The effect of varying UAV speed on the largest mission time

**Figure 13.** The average AoI and the largest mission time versus different $v_{max}$ and $v$.

To conclude, the above simulation results show that the proposed MUSM-CTPA algorithm can find the more optimal flight plans of $F$ and travel plan of the MUV cooperative than the benchmark algorithm such that the average AoI of all collected data is minimized. In addition, in the parameter setting of the MUSM-CTPA algorithm, $n$, $m$, $v$, $v_{max}$, $V_i$, $E_{max}$, $E_0$, and the presence of an MUV have different impacts on the average AoI. Among these parameters, $m$ and the presence of MUV have a greater impact, since multiple UAVs can perform tasks synchronously compared to a single UAV, while MUV can minimize the limitation of UAV energy. Therefore, it is necessary to consider the cooperation of multiple UAVs and a single MUV to assist data collection in WSN.

## 6. Conclusions

This paper investigates the multi-UAVs and single-MUV cooperative trajectory planning (MUSM-CTP) problem, which focuses on finding the flight trajectories of $m$ UAVs and the travel plan of the MUV cooperative. The objective of this problem is to minimize the largest mission time of $m$ UAVs such that the average AoI of all collected data is minimized. Then, we prove that the MUSM-CTP problem is NP-hard. Based on the definition of MUSM-CTP, we propose an algorithm named the multi-UAVs and single-MUV cooperative trajectory planning algorithm (MUSM-CTPA) to achieve cooperative trajectory planning. First, the minimum coverage algorithm based on affinity propagation (MCA-AP) is developed to find the hovering points set $C$ and the corresponding times set $T_C$ in a given WSN. Based on this result, the multi-UAVs trajectory planning algorithm based on the genetic

algorithm (MTPA-GA) is designed to find the flight trajectories of $m$ UAVs cooperatively visiting all the hover points, which are the positions and order of the hovering points $C_k$ visited by each UAV $f_k$. Finally, the scheduling strategy of the MUV is designed to find the optimal travel plan of the MUV. According to the massive simulations, we can verify that the proposed algorithm has great performance. In future work, we plan to consider the WSNs that utilize multiple UAVs and multiple MUVs for data collection and investigate the cooperative strategies for data collection and energy charging. Our goal is to further improve the efficiency and reliability of data collection to meet the demands of a wider range of practical applications.

**Author Contributions:** Conceptualization, Y.L. and Y.H.; methodology and data curation, Y.L.; validation, Y.L., Y.H. and C.L.; formal analysis, Y.L. and Y.H.; investigation, Y.L., Y.H., C.L. and D.L.; writing—original draft preparation, Y.L.; writing—review and editing, Y.L. and Y.H.; supervision, Y.H., C.L., Z.C. and D.L.; project administration, Y.H., C.L., Z.C. and D.L.; funding acquisition, Y.H. All authors have read and agreed to the published version of the manuscript.

**Funding:** This work was supported in part by the National Natural Science Foundation of China under Grant (62002022, 62202054) and the Fundamental Research Funds for the Central Universities (No. BLX201921, No. 2021ZY88).

**Institutional Review Board Statement:** Not applicable.

**Informed Consent Statement:** Not applicable.

**Data Availability Statement:** Not applicable.

**Conflicts of Interest:** The authors declare no conflicts of interest. The founding sponsors had no role in the design of the study; in the collection, analyses, or interpretation of data; in the writing of the manuscript; or in the decision to publish the results.

## Abbreviations

The following abbreviations are used in this manuscript:

| | |
|---|---|
| UAV | Unmanned Aerial Vehicle |
| WSN | Wireless Sensor Network |
| SN | Sensor Node |
| BS | Base Station |
| AoI | Age of Information |
| LOS | Line-of-Sight |
| MUV | Mobile Unmanned Vehicle |
| AP | Affinity Propagation |
| GA | Genetic Algorithm |
| MCA-AP | Minimum Coverage Algorithm based on Affinity Propagation |
| MTPA-GA | Multi-UAVs Trajectory Planning Algorithm based on Genetic Algorithm |
| MUSM-CTP | Multi-UAVs and Single-MUV Cooperative Trajectory Planning |
| MUSM-CTPA | Multi-UAVs and Single-MUV Cooperative Trajectory Planning Algorithm |

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
