# Peer review of "Optimization Algorithms for UAV-and-MUV Cooperative Data Collection in Wireless Sensor Networks"

_drones, doi:10.3390/drones7070408_

Round 1
Reviewer 1 Report
This paper studies the problem of planning the MUA and UAV path and flight trajectory, to collect data, the goal is to min the AoI and the constraints includes the energy. The usage of MUA to replace battery for the UAV is quite interesting and this topic has some practical impact. This paper is well organized. According to the simulation results of the authors, the performance of the proposed algorithm is promising.
There reviewer has the following suggestions for the authors.
* The introduces is lengthy and ordinary, it is hard for the reviewer to get the point what are the highlights of this paper. It is suggested to further summary the highlights.
* It is suggested to add a figure in Section 4 to introduce more clearly the relationship between the three phases.
* Please explain what is the reason phase two and three are not combined together?
* In Algorithm 3, multiple usage of keywords `optimal'. Are they appropriate?
* In the simulation, please add the MUA path into the resulting figure, to show how flight trajectory of UAV and path of MUA interact. Please also add the repower points.
The general quality of English language is fairly well. There are some minor issues of the this paper.
* A notation table is needed. Does h in line 174 have different meaning for h from Eq.(1)? Does U in Eq.(7) have different meaning from U in line 245?
* Line 299, `According to the' -> `By the' ?
Reviewer 2 Report
Comments to the Author:
In this paper, the authors studied a cooperative trajectory planning optimization problem for multiple UAVs and a single MUV in WSNs, which aims to minimize the average AoI with the constraints of UAV battery capacity. To solve the problem, they proposed an optimization algorithm on Affinity Propagation and Genetic Algorithm to find the flight trajectories of the UAVs and the travel plan of the MUV. And the authors provided extensive simulation results to verify the effectiveness of the algorithm. This work is well-organized, and I have the following comments:
1. In the motivation, the author introduced the issue of the battery replacement of the UAV in the long-term and energy-intensive missions. Why did you adopt the energy supplement from MUV rather than other methods like laser charging?
2. In the considered problem of the paper, data collection is performed when the UAVs hovers over the sensors. However, in some kinds of application scenarios, such as mountains and forests surveillance, there exist some obstacles blocking communication between the sensor and the UAV. Is there consideration for this issue in this paper and what is the solution?
3. In the related works, several articles solved the similar problems based on the algorithms of deep learning, e.g. reference [18]. Why didn’t you choose such solution idea as an option? Is the proposed algorithm more effective than those?
4. In the Subsection 3.3, there is only the power consumption model of the UAVs consumes on data collect. However, the sensors are also powered by the battery and their battery consumption costs on sensing, is there consideration of the energy consumption of the sensor?
5. In Section 4, to realize the sensor clustering and UAVs’ trajectory planning, the authors proposed AP-based algorithm and GA-based algorithm. What are the advantages of the ideas of AP and GA compared with other algorithms on clustering and path planning?
6. In Section 4, it is better to add more detailed description on the MUV's trajectory planning based on the greedy idea.
7. In Section 5, the authors compared the performance of the proposed algorithm with the greedy-based algorithm. The authors should add comparison with other algorithms to verify the effectiveness of the proposed algorithm.
The writting should be improved.
Reviewer 3 Report
- A Table for comparison with the related work should be conducted to emphasize the contributions of the manuscript.
- Some state-of-the-art studies related to UAV should be mentioned in the related work, DOI: 10.1109/ACCESS.2022.3173125, 10.3390/math11102376, 10.1109/JIOT.2022.3161571 - At the end of the equation, it should include a dot (if it is the end of the sentence) or a comma (if there is an explanation of the equation). Also, no "tab" space when the beginning of "where". - Remove "Definition" since it is not necessary.
- A Figure representing the flow should be provided since the proposed solution includes some steps.
- From lines 343 to 372, it can be used itemized or enumerated to represent the steps. Same for other explanations of the algorithm steps.
- Computational complexity of the algorithm is missing. Execution time evaluation is also missing.
- Why are the parameters of simulations selected? Are there any references?
- Comparison with state-of-the-art works should be conducted, not only changing environmental parameters.
- Additional bio-inspired algorithms could be compared, such as ACO, and PSO...
- At the end of the manuscript, the Abbreviations should be the list of the abbreviations used (e.g., UAVs: unmanned aerial vehicles).
- All references should include DOI for easy verification of the source.
- Major English revision should be conducted. Also, the format of the manuscript should be carefully revised.
- Some errors in the LaTex (lines 490, 492) (the word "and").
- If an abbreviation is already defined, it should use the abbreviation when it is mentioned later but not defined again. For example, Multi-UAVs and Single-MUV Cooperative Trajectory Planning (MUSM-CTP) repeats many times, instead, it should only use "MUSM-CTP" when later mentioned.
Round 2
Reviewer 2 Report
All my concerns have been addressed, so it can be accepted.
Author Response
Thank you for your reviewing.
Reviewer 3 Report
I have no further comments.
Author Response
Thank you for your reviewing.